# Inositol Phosphoryl Transferase, Ipt1, Is a Critical Determinant of Azole Resistance and Virulence Phenotypes in *Candida glabrata*

**DOI:** 10.3390/jof8070651

**Published:** 2022-06-21

**Authors:** Garima Shahi, Mohit Kumar, Nitesh Kumar Khandelwal, Atanu Banerjee, Parijat Sarkar, Sonam Kumari, Brooke D. Esquivel, Neeraj Chauhan, Amitabha Chattopadhyay, Theodore C. White, Naseem A. Gaur, Ashutosh Singh, Rajendra Prasad

**Affiliations:** 1Amity Institute of Biotechnology and Integrative Science and Health, Amity University Gurgaon, Gurgaon 122412, India; garimashahi47@yahoo.com (G.S.); mohitplawat007@gmail.com (M.K.); atanuj83@gmail.com (A.B.); 2Yeast Biofuel Group, International Centre for Genetic Engineering and Biotechnology, New Delhi 110067, India; sonamverma.sv85@gmail.com (S.K.); naseem@icgeb.res.in (N.A.G.); 3Department of Chemistry and Biochemistry, University of Arizona, Tucson, AZ 85721, USA; nitesh.k.khandelwal@gmail.com; 4CSIR-Centre for Cellular and Molecular Biology, Uppal Road, Hyderabad 500007, India; parijat@ccmb.res.in (P.S.); amit@ccmb.res.in (A.C.); 5School of Biological and Chemical Sciences, University of Missouri at Kansas City, Kansas City, MO 64110, USA; bdp3k9@mail.umkc.edu (B.D.E.); whitetc@umkc.edu (T.C.W.); 6Department of Microbiology, Biochemistry and Molecular Genetics, New Jersey Medical School, Rutgers, The State University of New Jersey, Newark, NJ 07103, USA; chauhan1@njms.rutgers.edu; 7Department of Biochemistry, University of Lucknow, Lucknow 226007, India

**Keywords:** *Candida glabrata*, sphingolipids, inositolphosphorylceramide, lipidomics, drug resistance, virulence

## Abstract

In this study, we have specifically blocked a key step of sphingolipid (SL) biosynthesis in *Candida glabrata* by disruption of the orthologs of ScIpt1 and ScSkn1. Based on their close homology with *S. cerevisiae* counterparts, the proteins are predicted to catalyze the addition of a phosphorylinositol group onto mannosyl inositolphosphoryl ceramide (MIPC) to form mannosyl diinositolphosphoryl ceramide (M(IP)_2_C), which accounts for the majority of complex SL structures in *S. cerevisiae* membranes. High throughput lipidome analysis confirmed the accumulation of MIPC structures in *ΔCgipt1* and *ΔCgskn1* cells, albeit to lesser extent in the latter. Noticeably, *ΔCgipt1* cells showed an increased susceptibility to azoles; however, *ΔCgskn1* cells showed no significant changes in the drug susceptibility profiles. Interestingly, the azole susceptible phenotype of *ΔCgipt1* cells seems to be independent of the ergosterol content. *ΔCgipt1* cells displayed altered lipid homeostasis, increased membrane fluidity as well as high diffusion of radiolabeled fluconazole (^3^H-FLC), which could together influence the azole susceptibility of *C. glabrata*. Furthermore, in vivo experiments also confirmed compromised virulence of the *ΔCgipt1* strain. Contrarily, specific functions of CgSkn1 remain unclear.

## 1. Introduction

Pathogenic fungi can develop clinical drug resistance after persistent drug exposure, which may impede successful treatment of human infections [1]. The most common human pathogenic fungi, *Candida albicans* and non *albicans Candida* (NAC) species, possess a repertoire of mechanisms to defy toxicity from multiple drugs. Among the various mechanisms of drug resistance, rapid drug efflux accomplished by select transporters stands out as the most prevalent mechanism of resistance [2]. Several factors have been identified that contribute to the fungal cellular responses to drugs, and antifungal resistance in *Candida* species is a complex multifactorial process.

Notably, several antifungal drugs target enzymes involved in lipid biosynthesis in *Candida* species. There is also a proven intricate relationship between membrane lipid homeostasis, micro-viscosity and drug resistance in *Candida* species [3,4]. For example, changes in membrane lipid phase and asymmetry affect azole resistance [5]. Any imbalances among membrane micro-domain (membrane raft) constituents lead to protein sorting defects and impacts on drug susceptibility in *C. albicans* [6]. Additionally, perturbation associated with phosphoglyceride (PGL) or sphingolipid (SL) metabolism also influences drug susceptibility, irrespective of whether a drug targets lipids or other cellular components [6,7]. Together, the drug susceptibility phenotype of *Candida* species appears to result from an orchestrated interplay between intracellular drug accumulation, drug efflux and the membrane lipid environment. The role of lipids is not restricted to drug resistance; and impacts a variety of cellular processes including biofilm formation and virulence [8,9]. Notably, certain SL structures are unique to fungi, which, apart from being novel drug targets, also appear to act as molecular signals, manifesting roles in diverse biological processes [10,11,12,13,14,15,16]. Independent observations from others and our group have revealed that in *C. albicans*, a complex relationship between SL and ergosterol homeostasis exists which has repercussions on virulence and susceptibility to antifungals [17,18,19,20,21]. For instance, an intermediate of SL metabolism, glucosylceramide, has been identified as a novel virulence factor in *C. albicans* and ceramide synthase as a virulence factor in encapsulated yeast *Cryptococcus neoformans* [22].

*C. glabrata* infection incidences stand out as the second most common NAC in hospitals [23], and reports state that *C. glabrata* is intrinsically less susceptible to azole antifungal with a mortality rate of 50% in patients with invasive infections [24,25,26]. Several studies have been done to understand the resistance mechanism of *C. glabrata* among which increased drug efflux due to upregulation of efflux pump is the main described mechanism, which is caused by gain-of-function (GoF) mutation in *CgPDR1,* a transcriptional regulator of drug resistance [27,28,29,30,31,32,33,34,35,36]. Higher adherence and high number of adhesins which are encoded by EPA family genes is crucial for the pathogenicity of *C. glabrata* as well [37]. Unlike other *Candida* species, *C. glabrata* cannot form true hypha so the phenotypic conversion to hyphal growth, which is an important virulence mechanism, is missing; however, *C. glabrata* has a completely different way of coping up with macrophage mediated phagocytosis. Following the engulfment by the macrophages, *C. glabrata* cells are somehow able to survive for a longer period of time and continue to divide [38].

In this study, we have specifically attempted to block the key synthesis step of SL metabolism by deleting the genes encoding CgIpt1 and CgSkn1, which based on their close homology with *S. cerevisiae* are predicted to be inositolphosphoryl transferases mediating the conversion of mannosyl inositolphosphoryl ceramide (MIPC) to mannosyl diinositolphosphoryl ceramide (M(IP)_2_C). We have elucidated the functionality of sphingolipids in *C. glabrata* by blocking the synthesis of M(IP)_2_C in *ΔCgipt1* cells leading to enhanced susceptibility to azoles.

## 2. Materials and Methods

### 2.1. Strains and Chemicals

*C. glabrata* strain BG2 referred to as wild type was gifted by Dr. Rupinder Kaur’s lab of fungal pathogenesis, CDFD, Hyderabad, India. Strains used in this study are listed in Appendix A. Yeast cultures were maintained in YPD broth (2% peptone, 1% yeast extract, 2% glucose) and YPD agar (YPD broth with 2% agar), unless otherwise stated. All the antifungal drugs used in this study were of analytical grade and were procured from Sigma Aldrich, Bangalore, India.

### 2.2. Deletion and Revertant Construction Strategy

Based on 18S rRNA sequence, *C. glabrata* is evolutionarily more closely related to the non-pathogenic yeast *S. cerevisiae* [39] than to the pathogenic yeast *C. albicans.* Based on the sequence homology of Ipt1 and Skn1 with *S. cerevisiae*, deletion mutants of *CgIpt1* and *CgSkn1* were constructed in a wild type (WT) haploid *C. glabrata* BG2 by employing NAT (Nourseothricin acetyltransferase) cassette as per our earlier established protocol [40]. The resulting knockouts were designated as *ΔCgipt1* and *ΔCgskn1*.

The revertant strain of *ΔCgipt1* was constructed by the episomal expression of the deleted gene. For revertant construction, a modified plasmid pGRB2.3_HphB was used, which was constructed by replacing the *URA3* gene with the HphB (for hygromycin resistance) selection marker. The GOI ORF was cloned in the pGRB2.3_HphB under its own promoter using Gibson assembly. Clones were verified by bacterial colony PCR and restriction digestion with specific enzymes. The resulting revertant was designated as *ΔCgipt1::IPT1.*

### 2.3. Lipid Extraction

Extractions of lipids were done as per Folch’s protocol described previously [41]. Briefly, an overnight culture was inoculated into 50 mL YPD broth to 0.2 OD_600_, grown for 6 h at 30 °C and was disrupted using glass beads. Supernatant was taken in a glass tube and lipids were extracted using a chloroform: methanol (2:1 *v*/*v*) ratio. Lipids extracted were dried with the help of an N_2_ flush and kept at −20 °C for further analysis. Dry weights of the extracted lipids were recorded at this point for the normalization of mass spectral data.

### 2.4. Mass Spectrometry Analysis

Lipid extracts were dissolved in 1 mL chloroform. To the 2–3 µL aliquots of these dissolved lipids, internal standards were added as described previously [42]. Lipid extracts were then analyzed on a Xevo TQ-XS triple quadrupole mass spectrometer (XEVO-TQS#WAA627, Waters, UK; Milford, MA, USA). Various lipid classes and individual molecular lipid species were determined using the neutral loss, negative and positive multiple precursor ion scans as described earlier [7,42,43]. Data processing was performed using the TargetLynx XS™ software (Waters, UK; Milford, MA, USA) and data was normalized to lipid dry weight and represented as nmol/mg lipid dry weight.

### 2.5. Gas Chromatography Mass Spectrometry (GCMS)

For free sterol analysis, base hydrolyzed lipid extract was derivatized with N, O-Bis (trimethylsilyl) trifluoroacetamide with trimethyl-chlorosilane (BSTFA/TMCS, Sigma, St. Louis, MO, USA) and analyzed on DB5-MS column (30 m × 0.2 mm × 0.20 μm) as described previously [43]. The retention time and mass spectral patterns of external standards were used for identification of sterol species.

### 2.6. Growth Assay

The growth kinetics was performed by a micro-cultivation method in a 96-well plate using a Liquid Handling System (Tecan, Austria) in YPD broth at 30 °C as described previously [40]. Briefly, overnight grown yeast cultures were diluted to 1.0 OD_600_ and 20 µL of each culture was mixed with 180 µL YPD broth in the 96-well plate. Different volumes of drugs, according to their indicated concentration, were added to the wells. The OD_600_ was measured at 30 min intervals up to 24 h.

### 2.7. Minimal Inhibitory Concentration (MIC) Measurements

The MICs for the strains were determined by the broth micro dilution method as described earlier [44]. Briefly, cells were grown for 12–14 h at 30 °C to obtain single colonies. They were then resuspended in a 0.9% normal saline solution to give an OD_600_ of 0.1. The cells were then diluted 100-fold in YPD medium. The diluted cell suspension was added to the wells of round-bottomed 96-well microtiter plates containing equal volumes of medium and different concentrations of drugs. A drug-free control was also included. The plates were then incubated at 30 °C for 24 h. The MIC test end point was evaluated by reading the OD_600_ in a microplate reader and was defined as the lowest drug concentration that gave 80% inhibition of growth compared with the growth of the drug-free control.

### 2.8. Spot Microdilution Assay

Samples of fivefold serial dilutions of each culture, each with cells suspended in normal saline to an OD_600_ of 0.1 were spotted onto YPD plates in the absence (control) or in the presence of drugs as described earlier [3]. Growth differences were recorded following incubation of the plates for 24 h at 30 °C.

### 2.9. Uptake Measurements of Radiolabelled ^3^H-FLC

The ^3^H-FLC accumulation assay was performed as described previously [45]. Overnight-grown (16 h shaking cultures in YPD medium at 30 °C) samples were washed three times with YNB (yeast nitrogen base), starved of glucose for 3 h, and then treated with ^3^H-FLC in YNB ± 2% glucose in technical duplicate. Samples were incubated with shaking at room temperature over the course of 3, 8 and 24 h. Previous azole import analyses have revealed that most fungal cells reach maximum intracellular drug accumulation by 24 h [46,47], so no measurements were taken after the 24 h time point. At each time point, the OD_595_ of each sample was taken, and then a stop solution, consisting of YNB + FLC (20 mM), was mixed with each sample. Then samples were poured over glass microfiber filters on a vacuum and washed again with 5 mL YNB. Filters with washed cells were placed in scintillation fluid and the radioactivity was measured in a Beckman Coulter LS 6500 Scintillation Counter. Values were adjusted to CPM per 10^8^ cells based on the OD_595_ of each sample recorded right before filtering.

### 2.10. Fluorescence Imaging and FRAP (Fluorescence Recovery after Photobleaching)

*C. glabrata* staining with *FAST*-DiI was performed as described previously [17]. *C. glabrata* culture was suspended at a density of 10^8^ cells/mL in 1 M sorbitol-0.1 M EDTA buffer and labelled using a final concentration of 10 µM *FAST*-DiI. Confocal imaging was carried out on an inverted Zeiss LSM 510 Meta confocal microscope using the 561 nm laser and fluorescence emission was collected using the 575–630 nm bandpass filter. Diffusion coefficient (D) and mobile fraction (M_f_) were calculated from quantitative FRAP experiments. FRAP experiments were performed with Gaussian spot-photobleaching and line-scanning mode with circular ROI of 1 µm radius. Data representing the mean fluorescence intensity in the membrane region within the bleach spot were corrected for background and analyzed. Non-linear curve fitting was used to analyze the fluorescence recovery plot and the graph was plotted using GraphPad Prism. Fluorescence recovery profiles and diffusion coefficients were analyzed as described by Koppel et al. [48].

### 2.11. Virulence Study

For virulence experiment, we used neutropenic murine model of systemic candidiasis [49]. To induce neutropenia, BALB/c mice were injected with 200 mg/kg intraperitoneal (ip) cyclophosphamide and subcutaneous cortisone acetate three days prior to the beginning of the experiment. *C. glabrata* WT, *∆Cgipt1* and *∆Cgipt1::IPT1* strains were grown in YPD broth at 30 °C overnight. Cells were then harvested, washed, and suspended in PBS to a required density. In BALB/c mice, strains were injected once via the lateral tail vein with 100 μL suspension containing 5 × 10^5^ *C. glabrata* cells. Mice were euthanized at 48 h post-infection. Kidneys were harvested and enumerated for *C. glabrata* burdens.

## 3. Results

### 3.1. The Deletion of CgIpt1 and CgSkn1 in Candida Glabrata

Based on the close homology with *S. cerevisiae*, the two ORFs, CAGL0G05313g and CAGL0I10054g of the SL biosynthetic pathway of *C. glabrata*, showed 51.51% and 61.76% sequence similarity in their protein sequences with ScIpt1 and ScSkn1, respectively. Both open reading frames (ORFs) were deleted individually in *C. glabrata* by employing a fusion-based PCR method and deletants were confirmed by semiquantitative PCR as described earlier [40]. Viable colonies obtained after single deletion of the *CgIpt1* and *CgSkn1* genes confirmed their non-essential nature. Both *ΔCgipt1* and *ΔCgskn1* mutant cells continued to grow on YPD liquid and solid media like wild type (WT) cells, implying that their deletion did not result in any change in cell fitness (data not shown).

### 3.2. Lipidomics of ΔCgipt1 and ΔCgskn1 Mutants Confirmed Their Involvement in SL Metabolism in C. glabrata

Inositol phosphoryl transferases have been shown to be the key enzyme in the formation of M(IP)_2_C, an abundant complex SL structure in the plasma membrane, reported in *S. cerevisiae* [50]. We analyzed the impact of its deletion in *C. glabrata* cells. For this we performed a comparative profiling of lipids extracted from *ΔCgipt1* and *ΔCgskn1* cells by employing high throughput MS-based lipidomic analysis to establish their role in SL metabolism and if its deletion has any repercussions on the overall lipid homeostasis. Our analysis detected all major PGL classes (lysophosphatidylcholine (LPC), phosphatidylcholine (PC), lysophosphatidylethanolamine (LPE), phosphatidylethanolamine (PE), Phosphatidylinositol (PI), phosphatidylserine (PS), phosphatidic acid (PA) and phosphatidylglycerol (PG)), sphingolipid class compositions (Inositolphosphorylceramide (IPC) and Mannosylinositolphosphorylceramide (MIPC)), ergosterol esters and neutral lipids (DAGs and TAGs).

There were no significant changes observed among the different PGL classes in *ΔCgipt1* and *ΔCgskn1* when compared to the WT (Figure 1A). Our analysis expectedly detected significantly higher amounts of total SLs (as much as 1.8-fold) in *ΔCgipt1* and *ΔCgskn1* cells (Figure 2A–D, Appendix A). The total content of ergosteryl esters also showed increased levels (as much as 12.8-fold) in *ΔCgipt1* and *ΔCgskn1* cells (Figure 2E,F, Appendix A). A 7.3-fold accumulation of the SL’s biosynthetic intermediate MIPC in *ΔCgipt1* and 1.6-fold accumulation of *ΔCgskn1* cells confirmed their role in SL metabolism as inositol phosphoryl transferases (Figure 2A–D).

We were able to detect 246 molecular lipid species spanning across 14 different classes of lipids and observed significant variations among several of them in *ΔCgipt1* and *ΔCgskn1* cells. We observed significant variations in PGL species contents of these mutants. For example, molecular species of LPC (LPC14:1 and LPC17:0), PC (PC26:1, PC26:0, PC28:1, PC28:0, PC30:2, PC31:1, PC33:2, PC33:1, PC33:0, PC36:1 and PC38:2), PE (PE26:0, PE33:2, PE34:3, PE35:2, PE35:1 and PE37:1) and PI30:1 species were seen to be reduced in *ΔCgipt1* and *ΔCgskn1* cells; while the level of a few species of PGL classes (PE36:2, PI36:2, PS30:0, PG32:2 and PG34:0) were in abundance in *ΔCgipt1* and *ΔCgskn1* cells (Figure 1B, Appendix A). The SL species IPC46:0–2, MIPC44:0–2, MIPC44:0–3, MIPC46:0–2 and MIPC46:0–3 were abundant in *ΔCgipt1* cells (Figure 2B,D).

### 3.3. CgSkn1 Does Not Significantly Impact MIPC Metabolism

The role of Skn1 in SL metabolism in yeasts remains unclear. Independent studies have reported that ScSkn1, a homologue of ScIpt1 in yeast, is involved in the biosynthesis of M(IP)_2_C. For instance, *Δ**Scskn1* and *ΔSc**ipt1* single and double deletion mutants of *S. cerevisiae* cells, when grown in a nutrient-rich medium, show a complete absence of M(IP)_2_C and thus confirm the role of ScSkn1 in MIPC metabolism [51]. Correspondingly, the other roles of Skn1 such as regulating β-1,6-glucan synthesis, hyphal and biofilm development, autophagy and virulence are also highlighted [50,52,53,54,55]. CaSkn1 role in the virulence of *C. albicans* is well demonstrated where its deletion along with another homologue, CaKre6, involved in glycan biosynthesis, results in attenuated virulence [54]. In the present study, we explored the role of Skn1 of *C. glabrata* in the biosynthesis of M(IP)_2_C. Our high throughput lipidomic analysis of *Δ**Cgskn1* cells and its comparison with *Δ**Cgipt1* cells showed that the deletion of the gene-encoding CgSkn1 did not cause any major changes in SL metabolism. Our data showed no significant changes in total SLs and its classes when compared with the WT cells, and in contrast to *Δ**Cgipt1* cells, relatively less accumulation of MIPC was detected in *Δ**Cgskn1* cells (Figure 2D). This further confirms that CgSkn1, unlike its counterpart in *S. cerevisiae*, may not be a major contributor in MIPC metabolism in *C. glabrata*; however, a supporting role is suspected.

### 3.4. ΔCgipt1 Cells Manifest Increased Drug Susceptibility

Both *ΔCgipt1* and *Δ**Cgskn1* cells were subjected to detailed drug susceptibility and phenotypic tests by employing three independent methods i.e., growth assay, minimal inhibitory concentration (MIC) and spot assays (Figure 3, Appendix A and Appendix A). Interestingly, we noticed that the accumulation of MIPC in *ΔCgipt1* cells was accompanied by raised susceptibility towards azoles (Figure 3A,B). *ΔCgipt1* showed increased susceptibility to both imidazoles (KTZ, MCZ and CTZ) and triazoles (FLC, ITR, PCZ) (Appendix A).

However, the Δ*Cgipt1* strain displayed no change in susceptibility towards a range of various other tested compounds like Congo red (CR), caffeine (CAF), calcofluor white (CFW), cycloheximide (CHX), verapamil (VER), o-phenanthroline (OP), 4-nitroquinoline (4NQO), chloramphenicol (CHL), acetic acid (AcOH; pH 4.5), anisomycin (ANM), naftifine (NAFT), myriocin (MYR), terbinafine (TRB) and hydrogen peroxide (H_2_O_2_) (Appendix A). The growth of Δ*Cgipt1* cells in comparison to parent cells on solid agar media also remain unaffected at different temperatures, pHs, carbon sources and iron chelators (Appendix A).

In contrast to *ΔCgipt**1* cells, the deletant *Δ**Cgskn1* did not manifest any change in drug susceptibility (Appendix A). However, the double deletion mutant, *Cgipt1*/*Cgskn1ΔΔ,* expectedly showed increased susceptibility towards azoles caused by the absence of functional CgIpt1 in *Cgipt1*/*Cgskn1ΔΔ* cells (Figure 3C). Since *Δ**Cgskn1* cells did not display significant changes in SL metabolism and in drug susceptibility, we did not include it in our further experiments.

### 3.5. ΔCgipt1 Cells Revealed Increase Levels of Sterols

From our lipidomic analyses, we recorded an increase in the total ergosteryl esters as well as their species in *ΔCgipt1* cells (Figure 2E,F). Following this we checked the total sterol and their species content in *ΔCgipt1* by employing GC-MS. Our GC-MS analysis could detect eight different sterol species, namely squalene, dehydroergosterol, ergosterol, fecosterol, episterol, fungisterol, lanosterol and UI-sterol. The content of total sterol and their intermediates was significantly higher in *ΔCgipt1* cells compared to WT cells (Figure 4A,B). Among these, ergosterol and squalene were the most abundant lipids detected. It appears that *ΔCgipt1* cells tend to accumulate free sterols in response to the depletion of M(IP)_2_C as a compensatory mechanism for the loss of complex SLs (See Section 4).

### 3.6. Deletion of CgIpt1 Leads to Reduced PM Rigidity

Since drug diffusion and susceptibility of *Candida* cells has been linked to the physical state of the PM earlier [3,17,56,57], we explored if changes in MIPC levels observed in *ΔCgipt1* cells could alter the permeability of the PM. For measuring membrane fluidity changes in *ΔCgipt1* cells, we employed FRAP analysis by using the dye *FAST*-DiI as described in Materials and Methods. Both WT and *ΔCgipt1* cells were photobleached and the region was imaged over time to check the recovery of the dye fluorescence. Figure 5A depicts the representative fluorescence recovery experiment images of WT and *ΔCgipt1* strains. As shown in the overlapping recovery plot, *ΔCgipt1* cells showed faster recovery of the dye fluorescence to the bleached area as compared to the WT strain (Figure 5B).

Analysis of fluorescence recovery plots showed that *ΔCgipt1* mutant cells indeed showed a higher diffusion coefficient value (3.84 × 10^−9^ cm^2^/s) as compared to the WT strain that showed a lower diffusion coefficient value (0.84 × 10^−9^ cm^2^/s) (Figure 5C). Further, we calculated the mobile fraction as described in Materials and Methods. Along with higher diffusion coefficient values, we also observed higher mobile fraction of the dye in *ΔCgipt1* cells (58.35%) when compared with WT cells (40.74%) (Figure 5D). These results confirmed a higher extent of recovery (mobile fraction) and rate of recovery (diffusion coefficient) in *ΔCgipt1* cells relative to WT cells. Together, we conclude that M(IP)_2_C in *ΔCgipt1* cells are essential to maintain optimum viscosity of PM and its absence tends to make the PM more fluid.

Changes in lipid ratio compositions can be a good indicator of alterations in membrane fluidity. A closer look at the PGL molecular species showed significant changes in lipid ratios. For instance, the PC34:2/PC34:1 and PC36:2/PC36:1 ratio was higher in *ΔCgipt1* when compared with WT (Figure 1C). Increase in unsaturation index of these lipid ratios correlates with loss of plasma membrane rigidity in *ΔCgipt1* cells (discussed later). It is reported that imbalances in PE levels influence the viscosity of the plasma membrane bilayer where an increase in PE content leads to increased rigidity of the membrane [58]. In the present case, *ΔCgipt1* cells show reduced ratios of PE34:3/34:1 and PE34:3/34:2 (Figure 1C) which implies that the reduction of PE species as well as their lipid ratios in *ΔCgipt1* cells contribute to increased plasma membrane fluidity in these cells. Of note, the amounts of the individual PGL species might not always change between the WT and the deletion strains, however, significant variations could be still observed upon comparing the PGL species ratios, effecting overall membrane homeostasis. Total contents of saturated FA containing PGLs and odd chain FA containing PGLs were significantly reduced in *ΔCgipt1* cells as compared to WT cells (Appendix A). The principal component analysis (PCA) could further validate the statistically significant changes in molecular lipid species between WT and *ΔCgipt1* cells (Appendix A). No significant change was seen in the total diacylglycerol (DAG) and triacylglycerol (TAG) contents in *ΔCgipt1* (Appendix A) strain.

### 3.7. ΔCgipt1 Cells Show Enhanced Diffusion of ^3^H-FLC

It is now well established that entry of azoles into *Candida* species and other pathogenic fungi is predominantly through facilitated diffusion. Increased antifungal drug uptake and accumulation by fungi could lead to increased drug susceptibility, while reduced drug accumulation could result in resistance to the drug [46,47,59,60]. We investigated whether increased FLC diffusion/accumulation could explain the increased fluconazole susceptibility of *ΔCgipt1* cells. We monitored FLC entry by measuring the intracellular accumulation of radiolabeled fluconazole (^3^H-FLC) over time as explained earlier [45]. Briefly, *ΔCgipt1* and WT cells were treated with ^3^H-FLC with and cell samples were withdrawn at the indicated time interval and the intracellular FLC was measured. The assay was performed in both energized (2% glucose) and deenergized (glucose starved) conditions to gain an indication of drug accumulation in actively dividing cells, as well as less active, more stationary cells, respectively.

In both energized and deenergized conditions, the final intracellular FLC concentration was higher in the *ΔCgipt1* strain (Figure 5E). In energized conditions, the *ΔCgipt1* strain accumulated significantly more fluconazole as early as 3 h post-treatment and continued to accumulate significantly more FLC compared to WT over the course of 24 h. In deenergized conditions, the *ΔCgipt1* strain accumulated higher fluconazole than the WT strain by the 8-h timepoint, but did not have significantly different intracellular FLC from WT until the 24-h post-treatment time point.

### 3.8. ΔCgipt1 Cells Show Attenuated Virulence in Mouse Model

To assess whether the deletion of CgIpt1 alters virulence, we used the neutropenic murine model of systemic candidiasis [49]. Neutropenic BALB/c mice were injected once used to assay kidney fungal burden by injecting them with WT, *ΔCgipt1* and *ΔCgipt1::IPT1* strains as described in Materials and Methods. Colonization (CFU/g) of kidneys for each strain at 24 to 48 h post-infection was determined by routine CFU determination procedures. Interestingly, the *ΔCgipt1* strain showed a significant decrease in CFU counts compared to the WT parental strain at 48 h post-infection (Figure 6). The *ΔCgipt1::IPT1* showed recovery of CFU counts at 48 h post-infection (Figure 6). Reduction of kidney burden in the mouse model suggests a role for CgIpt1 in the virulence and pathogenicity mechanisms of *C. glabrata.*

## 4. Discussion

Past studies have underscored the relevance of SLs in influencing physiological processes in yeast cells. Not only do these molecules provide signaling cues to impact virulence of *C. albicans* cells, but their imbalanced lipid homeostasis also affects membrane protein cellular trafficking and drug susceptibility. The inherent structural peculiarities of acidic SLs in yeast (the absence of inositolphosporylceramides in mammals) make them novel drug targets, and hence justifiably continue to draw research attention. The present study has explored the hitherto unknown nature of inositolphosporylceramides metabolism in *C. glabrata*, which is the second leading cause of human fungal infection after *C. albicans*. We selected two ORFs, CAGL0G05313g (CgIpt1) and CAGL0I10054g (CgSkn1), from the *C. glabrata* genome, which, based on their close homology with *S. cerevisiae* counterparts, are predicted to encode for inositol phosphoryl transferases. The successful single (*ΔCgipt1* and *ΔCgskn1*) and double deletion (*Cgipt1*/*Cgskn1ΔΔ*) of *Cg**Ipt1* and *CgSkn1* confirmed the non-essential nature of the two homologues which contrasts with their indispensability in the budding yeast *S. cerevisiae*. The deletion of *ΔCgipt1* or *ΔCgskn1* in *C. glabrata* did not impact cellular growth of the cells implying that the loss of either of the two genes did not impose any fitness cost.

High throughput lipidomics of the parental strain compared to the two single-deletant strains (*ΔCgipt1* and *ΔCgskn1*) revealed interesting insights into the physiological relevance of CgIpt1 and CgSkn1. The quantitation of IPC intermediate levels indicated that the ORF putatively encoding CgIpt1 is a major player of inositolphosporylceramides transferase activity, wherein CgSkn1 probably has a minor role in IPC metabolism in *C. glabrata* cells. This is well supported by the fact that *ΔCgipt1* cells exhibited a strong phenotype reflected in an increased drug susceptibility to tested azoles while the susceptibility of *ΔCgskn1* cells towards azoles remained unchanged from the parental strain. The dominance of CgIpt1 in influencing drug susceptibility was further evident by increased susceptibility towards azoles observed in the *Cgipt1*/*Cgskn1ΔΔ* strain compared to the *ΔCgskn1* strain.

The role of yeast SL genes has been established in fundamental pathways like endocytosis, GPI-anchored proteins and vesicular trafficking, which are required for cell wall synthesis where the mutant genes were observed to show increased susceptibility on cell wall perturbing agents such as calcofluor white and SDS along with specific alteration in cell wall properties [61,62,63,64]. In our finding, the deletion of *CgIpt1* did not impact the cell wall significantly (Appendix A). The fact that *ΔCgipt1* cells have a structurally and functionally intact cell wall and yet present with a compromised plasma membrane is an interesting observation but is not surprising. One can argue that certain yeasts may have a cell wall defect but show no change in plasma membrane permeability. Furthermore, even if a cell shows resistance to cell wall perturbing agents, the overall cell permeability could still be altered [65].

Notably, the content of sterol and its intermediates were significantly higher in the *ΔCgipt1* strain compared to the parental strain. While an increase in ergosterol levels is commonly linked to enhanced resistance towards azoles in yeasts, this was not the case in the *C. glabrata IPT1* mutant. Even with a high level of ergosterol in *ΔCgipt1*, the mutant showed increased susceptibility to azoles, implying that imbalances in SLs alone can influence drug susceptibility in *C. glabrata*. This does not fit the usual convenient notion where azole susceptibility can be easily pinned on low ergosterol content and accumulation of branched “toxic” sterol intermediates [66]. Considering the fact that *ΔCgipt1* cells are trying to compensate for the loss of M(IP)*_2_*C, by subtle changes in other lipid species composition, one could speculate that since M(IP)*_2_*C structures in fungal membranes are mostly composed of hydroxylated very-long-chain fatty acid (VLCFA) [67], these structures can directly affect the order of the membrane. In this light, depletion of M(IP)*_2_*C structures in the *ΔCgipt1* cell membrane could result in reduced membrane order, which in turn could be compensated by an increase in the content of specific sterol structures. In fact, in a specific study in *S. cerevisiae*, it was shown that *IPT1* deletion strongly affects the rigidity of gel domains without influencing their relative abundance, whereas no significant alterations could be perceived in ergosterol-enriched domains [68]. In a separate study in *C. albicans*, we have shown that deletion of *IPT1* destabilizes the membrane microdomains which in turn impairs the functionality of Cdr1, which is a major fluconazole efflux pump [20]. Therefore, it would not be wrong to presume that deletant of *IPT1* in *C. glabrata*: accumulates MIPC in the membrane, has low M(IP)*_2_*C which is balanced by increase in sterol structures, local membrane micro-environment is altered in such a way that promotes higher drug diffusion and drug efflux pumps are thereby rendered defunct, leading to increased azole susceptibility of *ΔCgipt1* cells. Although specific sterol structures and altered membrane microdomain could directly affect the functionality of the protein localized therein [69], some of which may be directly linked to azole resistance. However, another study in *C. albicans* cells described a different mechanism for azole susceptibility [70]. Zhang et al. argue that the reduction of ergosterol upon fluconazole treatment of *C. albicans* impairs vacuolar acidification, whereas concomitant ergosterol feeding restored V-ATPase function and cell growth. In *C. glabrata* there is a possibility that the plasma membrane of *ΔCgipt1* cells accumulates certain sterol structures that disrupt the membrane order, or there may be some other mechanism could be involved. There are also reports on *C. glabrata* which do not find a direct correlation between increased sterols and enhanced susceptibility to fluconazole [71,72,73]. No major differences were noted in the level of expression of *ERG* genes between drug resistant and susceptible isolates [74,75], implying no direct link between increased azole resistance and overexpression of *CgERG* genes [76].

It was observed that the lipid imbalances caused by the *CgIpt1* gene deletion led to an increase in membrane fluidity (revealed by FRAP measurements). This was supported by increased drug uptake (indicated by increased intracellular ^3^H-FLC accumulation) in the *ΔCgipt1* strain compared to its parent WT strain. Increased membrane fluidity, allowing more drug into the cells, could contribute to the increased drug susceptibility (revealed by broth drug microdilution) of the *ΔCgipt1* strain (Figure 7). Together, we report the first functional characterization of two putative inositolphosporyltransferase-encoding ORFs of *C. glabrata* genome. We show that among the two homologs, CAGL0G05313g (CgIpt1) is a major inositolphosporyltransferase which influences drug susceptibility and virulence of *C. glabrata*, while its close homologue CAGL0I10054g (CgSkn1) does not appear to be a major contributor of IPC metabolism and its functional relevance remains to be explored.

## Figures and Tables

**Figure 1 jof-08-00651-f001:**
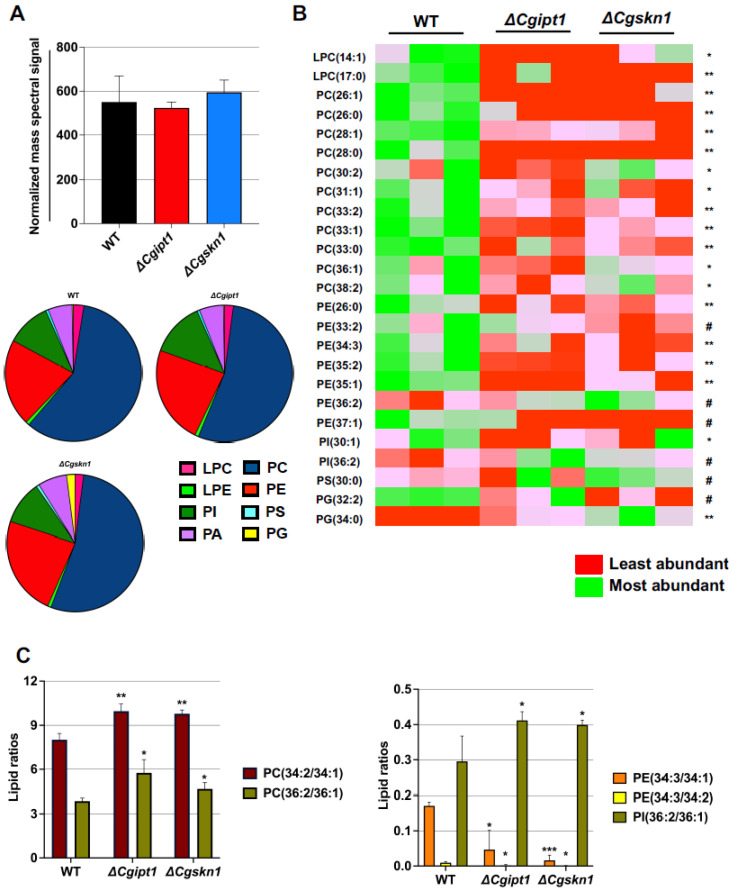
Lipidomic analysis reveals significant changes in PGL species in *ΔCgipt1* and *ΔCgskn1* mutants of *C. glabrata*. (**A**) The total PGL content, along with the content of specific PGL classes of PC, LPC, PE, LPE, PI, PS, PG and PA, in WT, *ΔCgipt1* and *ΔCgskn1* mutants of *C. glabrata* are depicted. (**B**) Compared to the WT, *ΔCgipt1* and *ΔCgskn1* mutants of *C. glabrata* show significant changes in PGL molecular species compositions. The PGL species are represented as “total number of carbons in the acyl chains: total number of carbon-carbon double bonds in the acyl chains”. Data represents nmol per mg lipid dry wt. as total normalized mass spectral signal and can be found in Appendix A. Mean ± SD of three replicates is plotted. Significant changes in PGL species with a *p*-value of <0.05 are represented by * (WT versus *ΔCgipt1*), # (WT versus *ΔCgskn1*), and ** (WT versus *ΔCgipt1* and *ΔCgskn1*). (**C**) Variations in the ratio of lipid species between WT vs. *ΔCgipt1* and WT vs. *ΔCgskn1* are shown. Data are represented as mean ± SEM of three independent replicates. Significant differences from WT ratios with a * *p* value ≤ 0.05; ** *p* value ≤ 0.006; *** *p* value ≤ 0.0001 calculated using unpaired student’s *t*-test.

**Figure 2 jof-08-00651-f002:**
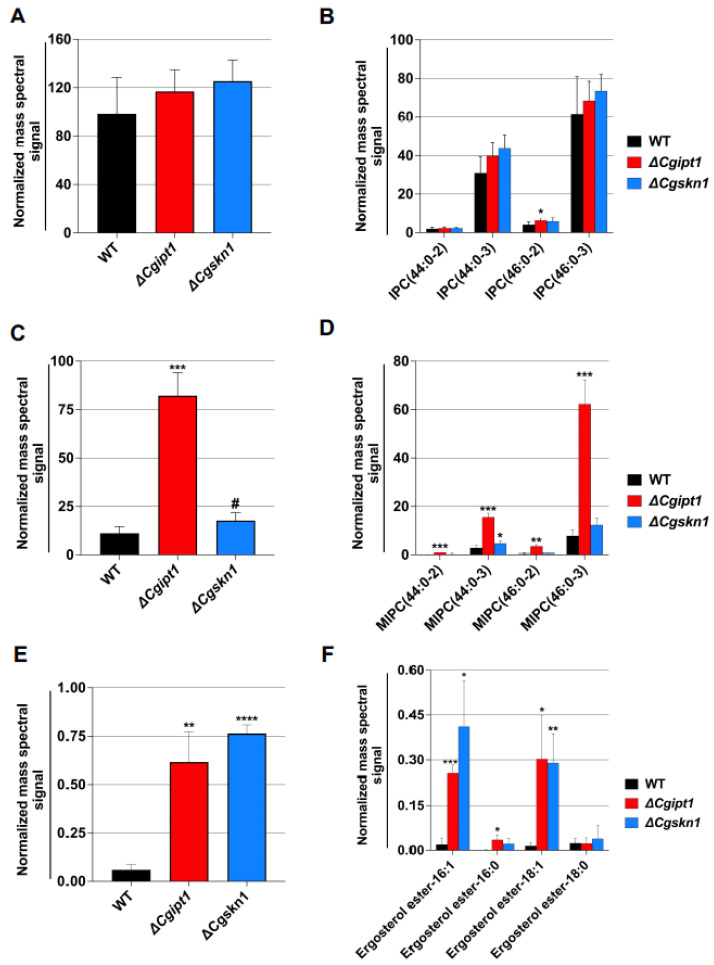
Alterations in SL and ergosteryl ester contents *ΔCgipt1* and *ΔCgskn1* mutants of *C. glabrata*. (**A**) Total IPC content. (**B**) IPC species content. (**C**) Total MIPC content. (**D**) MIPC species content. (**E**) Total ergosteryl ester content. (**F**) Ergosteryl ester species content. IPC and MIPC species are represented as “total number of carbons in the sphingoid base and acyl chains: total number of carbon-carbon double bonds in the sphingoid base and acyl chains- number of hydroxyl groups present in the sphingoid base and acyl chains”. Data represents nmol per mg lipid dry wt as total normalized mass spectral signal and can be found in Appendix A. Mean ± SD of three replicates is plotted and compared to WT. * *p* value ≤ 0.05; ** *p* value ≤ 0.007; *** *p* value ≤ 0.0008; **** *p* value < 0.0001; # *p* value 0.10 were calculated using unpaired student’s *t* test.

**Figure 3 jof-08-00651-f003:**
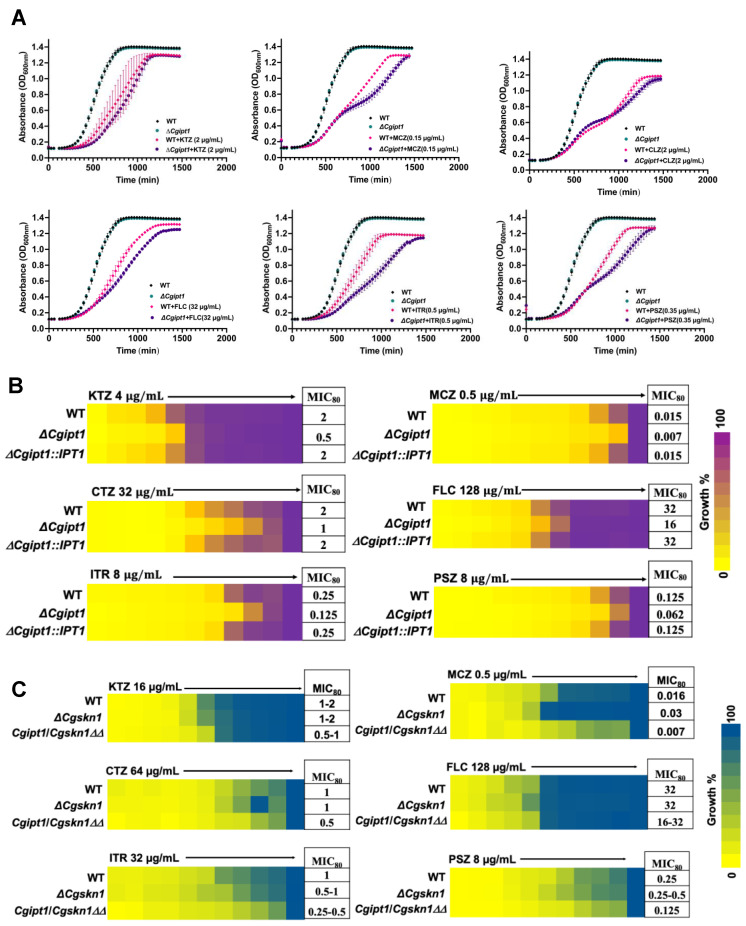
Drug susceptibility analysis of *ΔCgipt1*, *ΔCgskn1* and *Cgipt1*/*Cgskn1 ΔΔ* mutants of *C. glabrata*. Drug susceptibility to KTZ, MCZ, CTZ, FLC, ITR and PCZ was determined by (**A**) growth curve analysis, (**B**,**C**) broth microdilution assay as described in our earlier publications and briefly mentioned in Section 2.

**Figure 4 jof-08-00651-f004:**
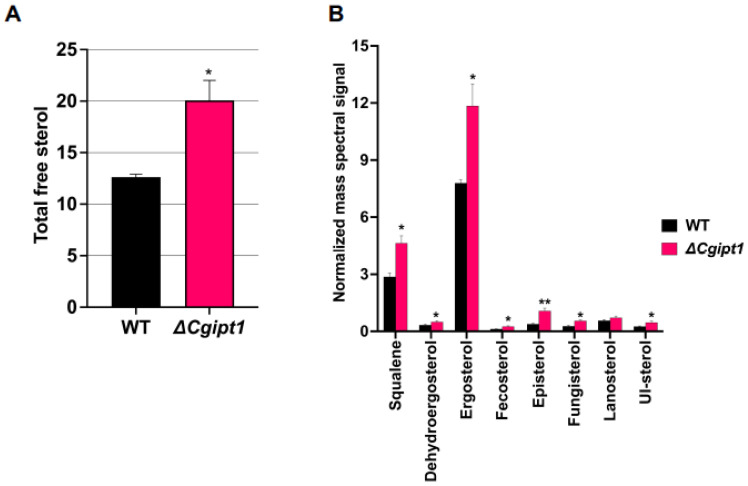
Deletion of CgIpt1 perturbs the sterol homeostasis in *C. glabrata*. (**A**) Total free sterol content. (**B**) Content of different sterol species detected by GCMS. Mean ± SD of five replicates is plotted. * *p*-values < 0.05; ** *p*-values < 0.009 calculated using unpaired student’s *t* test.

**Figure 5 jof-08-00651-f005:**
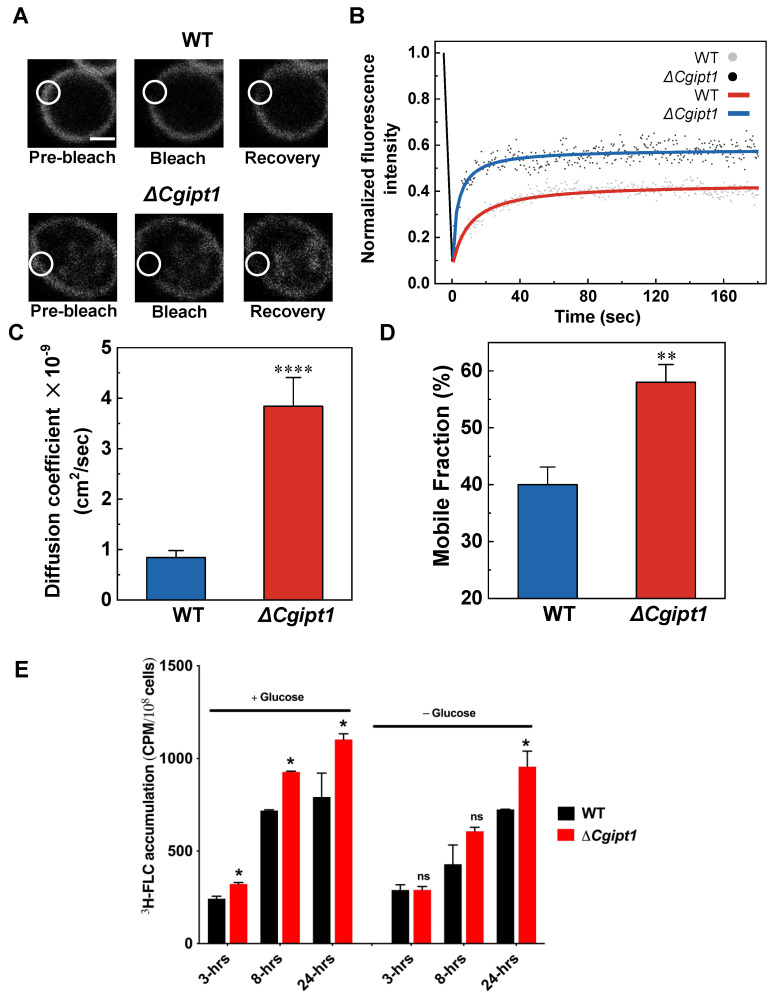
Disruption of CgIpt1 causes reduced PM rigidity. (**A**) Representative images of *FAST*-DiI dye-labeled plasma membrane of WT and *ΔCgipt1* cells (left column, t = 0 s, pre-bleach). A region of interest (ROI, white circle) was photobleached and cells were imaged immediately thereafter (second column, ‘bleach’) and after 180 s post-bleach (third column, ‘recovery’). (**B**) Overlapped fluorescence recovery curves showing faster recovery in *ΔCgipt1* cells relative to WT cells. (**C**) Diffusion coefficients (**D**) and mobile fraction (M_f_) were calculated for WT and *ΔCgipt1* cells. Experiments were performed in biological triplicates and statistical significance values (*p* < 0.0001 (****); *p* < 0.0014 (**)) were calculated using unpaired Student’s *t*-test. (**E**) ^3^H-FLC accumulation in the WT and *ΔCgipt1* strains in energized (+Glucose) and deenergized (−Glucose) conditions. Significant differences in FLC accumulation between WT and *ΔCgipt1* were calculated and the *p*-values of <0.05 are represented by *; ns, not significantly different.

**Figure 6 jof-08-00651-f006:**
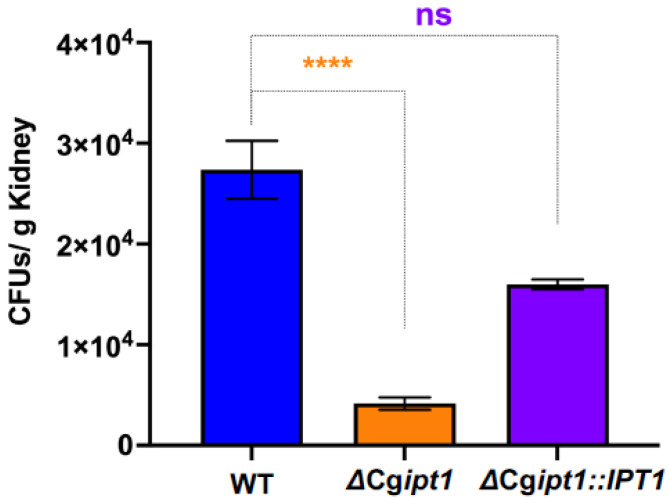
*Δ**Cgipt1* cells show reduced fungal burden in vivo. *ΔCgipt1* cells show reduced fungal burden in kidney. CFU recovered from kidney were significantly low in *ΔCgipt1* cells as compared with the WT cells as mentioned in the indicated time points. Significant differences at CFU levels are indicated as **** *p*< 0.0001; ns, not significantly different calculated using Student’s *t*-test on graph pad prism 9.

**Figure 7 jof-08-00651-f007:**
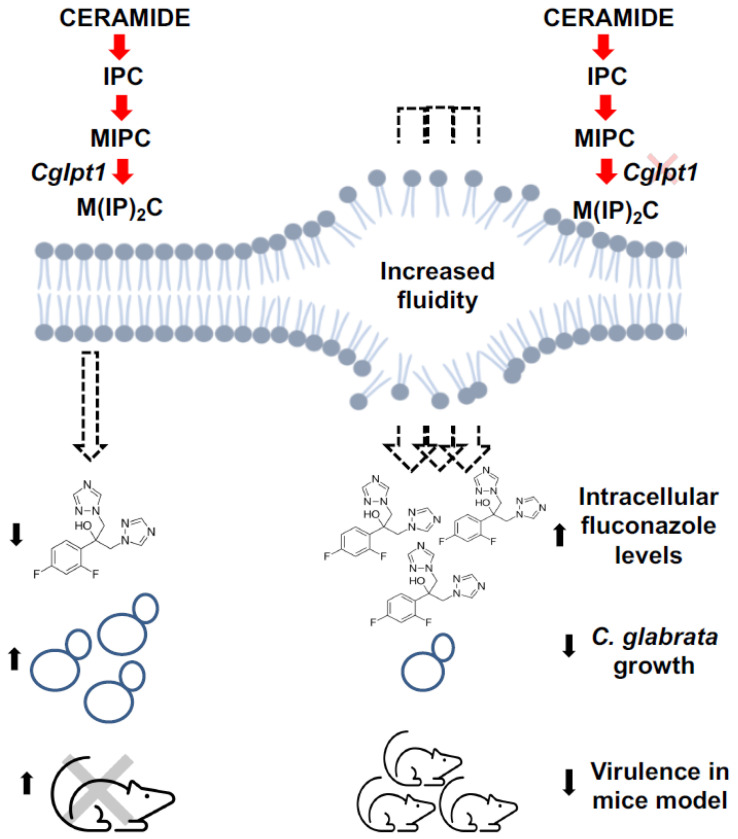
Model depicting the impact of *CgIpt1* deletion on *C. glabrata* cells. The deletion of the gene results in imbalances in SL homeostasis and consequently an increased membrane fluidity, which further leads to an enhanced diffusion of the antifungal FLC. The resulting increased intracellular FLC increases FLC susceptibility, impedes cellular growth and compromises the virulence of *C. glabrata* in a mouse model.

## Data Availability

Not applicable.

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
