# Peer review of "Inositol Phosphoryl Transferase, Ipt1, Is a Critical Determinant of Azole Resistance and Virulence Phenotypes in Candida glabrata"

_jof, 2022, doi:10.3390/jof8070651_

Round 1
Reviewer 1 Report
The manuscript by Shahi et al advances, in a convincing manner, the functional analysis of two genes in C. glabrata encoding inositol phosphoryl transferases. The paper is well written and the results are interesting and move forward aspects of relevance for the understanding of important aspects of the physiology of C. glabrata, with impact in virulence and in resistance to antimicrobials. I have one major scientific issue that in my view is not properly addressed, and a few that are minor aspects.
1. The most important issue that I think has not been properly addressed concerning the impact of the deletions in the azole-tolerance phenotype and the subsequent link with the lipidomic analyses. The comparison of the effect of the two gene deletions has been performed in the absence of any azole and in these conditions the cells show higher levels of ergosterol which would prompt them to be more tolerant (as also recognized by the authors). However, this was not the case and this is something that the authors address poorly and that, in my view, considering the argument of involvement of this pathway in azole resistance shouldn't be overlooked (and in the current version of the paper it seems so). Since the authors have access to the technology it would be of most importance to see what happens to the distribution of sterols in the PM of these mutants IN THE PRESENCE OF AZOLES since one reason for them to be more sensitive could be a higher accumulation of the toxic sterol forms. Since the phenotype is tested in the presence of the azole it only makes sense that the association with the lipid composition is performed in the same conditions. This will disclose an important feature to understand how different lipid compositions of the PM can actually impact azole tolerance.
Minor aspects
1. Saying that knowledge about virulence and antifungal resistance in C. glabrata remains largely unknown seems a bit of an excess considering so much that is done on the field (e.g. Cg resistance in isolates is largely determined by the acquisition of gain-of-function mutations in CgPdr1, and that can't/shouldn't be ignored!);
2. The phenotype of Acetic acid is tested at what pH? Being an organic acid it is fundamental to have the pH monitored as otherwise the authors are not observing a real effect of Acetic acid that relies on the properties of its undissociated form;
3. The growth curve of WT in the presence of CLT shows almost no growth (therefore no growth could also be expected from the mutant). What is the usefulness of showing this growth curve if not even the WT grows?
4. In the chart describing the differences in FLC accumulation some statistical analysis seem not coherent (how come the last point at 24h with that error bar has more significance than the previous one that has a considerable higher differences). Revision of this should be performed. Also, the differences in some of these points appear minor and we don't know how they translate into concentrations (meaning that in some cases the difference might not actually represent a meaningful difference of concentrations inside the cells). In my view the authors can make a better discussion of these results because it is merely mentioned that the mutant accumulates more than the wt but little is discussed concerning the magnitude (small) of the differences and their impact in the overall physiology of the cells.
Reviewer 2 Report
This paper attempts to characterize two C. glabrata genes involved in lipid synthesis. The paper is well written, experiments are adequate and address the aims of the paper. Some improvements are required however (see below, in no particular order of importance):
1- In the introduction the authors state "SL are unique to fungi..." Obviously other organisms also contain sphingolipids so this statement has to be modified.
2-The authors generated a revertant strain for IPT1, however it does not seem to have been included for most experiments. The revertant needs to be included in all experiments to support the data. Furthermore the authors do not seem to have generated a revertant for SKN1. This needs to be performed since SKN1 was excluded from many of the experiments based on mutant phenotype, but to begin with the mutant data cannot be trusted without a revertant.
3-The authors set to investigate the role of 2 genes involved in lipid biosynthesis. However based on the data in earlier experiments they decided to exclude one deletion from further analysis. I believe the skn1 mutants should not be excluded from the remaining experiments such as virulence, cell surface rigidity, ergosterol content etc.. The skn1 mutant might still exhibit interesting phenotypes that might help in characterizing it.
4-In addition to organ fungal burden load, a mouse survival experiment should be carried out by injecting the fungus in the tail vein and recording morbidity vs time. This is a better indicator of virulence.
5-It is a bit surprising that even though the deletions affected membrane fluidity and architecture resulting in increased susceptibility to antifungal drugs, there was absolutely no difference between the mutant and the WT in a spotting assay to chemicals that specifically target the cell wall and the plasma membrane (fig S1B). There are numerous articles in the literature that correlated membrane disturbance with increased susceptibility to such agents. The discussion should address this issue.
Reviewer 3 Report
Dear Authors,
I read your manuscript with interest, especially to the direct connections with new targets in this period of Multidrug resistance pathogens and fungi in specific. The material and methods are good and well-presented, results are in line with the methodology. I report some notes to improve your paper:
1) In vivo in italic style, check the text
2) the introduction is approximate, with general and unrelated notions. it needs to be reviewed and several points need to be investigated. Furthermore, lines 72-77 are found and should be removed from the introduction
3) why did you not use a standard control stain of C. glabrata?
4) Line 321-322 format, please
5) Figure 1b in supplementary is not readable and no ranges are visible.
Round 2
Reviewer 2 Report
The authors responded to my queries. I am satisfied with the responses